# Metabolomic Profile of Abdominal Aortic Aneurysm

**DOI:** 10.3390/metabo11080555

**Published:** 2021-08-22

**Authors:** Jüri Lieberg, Anders Wanhainen, Aigar Ottas, Mare Vähi, Mihkel Zilmer, Ursel Soomets, Martin Björck, Jaak Kals

**Affiliations:** 1Department of Surgery, Institute of Clinical Medicine, University of Tartu, 8 Puusepa Street, 51014 Tartu, Estonia; Jyri.Lieberg@kliinikum.ee; 2Department of Vascular Surgery, Tartu University Hospital, 8 Puusepa Street, 51014 Tartu, Estonia; 3Department of Surgical Sciences, Section of Vascular Surgery, Uppsala University, SE-751 85 Uppsala, Sweden; Anders.Wanhainen@surgsci.uu.se (A.W.); martin.bjorck@surgsci.uu.se (M.B.); 4Department of Biochemistry, Institute of Biomedicine and Translational Medicine, Centre of Excellence for Genomics and Translational Medicine, University of Tartu, 19 Ravila Street, 50411 Tartu, Estonia; Aigar.Ottas@ut.ee (A.O.); Mihkel.Zilmer@ut.ee (M.Z.); Ursel.Soomets@ut.ee (U.S.); 5Institute of Mathematics and Statistics, University of Tartu, 18 Narva mnt. Street, 51009 Tartu, Estonia; Mare.Vahi@ut.ee

**Keywords:** abdominal aortic aneurysm, biomarkers, metabolites, pathophysiology

## Abstract

Abdominal aortic aneurysm (AAA) is characterized by structural deterioration of the aortic wall, leading to aortic dilation and rupture. The aim was to compare 183 low molecular weight metabolites in AAA patients and aorta-healthy controls and to explore if low molecular weight metabolites are linked to AAA growth. Blood samples were collected from male AAA patients with fast (mean 3.3 mm/year; range 1.3–9.4 mm/year; *n* = 39) and slow growth (0.2 mm/year; range −2.6–1.1 mm/year; *n* = 40), and from controls with non-aneurysmal aortas (*n* = 79). Targeted analysis of 183 metabolites in plasma was performed with Absolute*IDQ* p180 kit. The samples were measured on a QTRAP 4500 coupled to an Agilent 1260 series HPLC. The levels of only four amino acids (histidine, asparagine, leucine, isoleucine) and four phosphatidylcholines (PC.ae.C34.3, PC.aa.C34.2, PC.ae.C38.0, lysoPC.a.C18.2) were found to be significantly lower (*p* < 0.05) after adjustment for confounders among the AAA patients compared with the controls. There were no differences in the metabolites distinguishing the AAA patients with slow or fast growth from the controls, or distinguishing the patients with slow growth from those with fast growth. The current study describes novel significant alterations in amino acids and phosphatidylcholines metabolism associated with AAA occurrence, but no associations were found with AAA growth rate.

## 1. Introduction

Abdominal aortic aneurysm (AAA) is characterized by structural deterioration of the aortic wall, leading to aortic dilation, growth, and rupture [1]. Infra-renal AAA is defined as a maximum infra-renal abdominal aortic diameter of ≥3 cm. In addition to AAA rupture with a mortality over 80% [2,3], patients with AAA often have other cardiovascular diseases [4]. Recent systematic review and meta-analysis suggests that, besides timely repair of AAA, secondary cardiovascular prevention (e.g., statins) is also very important to improve prognosis in these patients [5]. 

AAA is mostly an asymptomatic but potentially fatal condition because progressive enlargement of the abdominal aorta is spontaneously evolving towards rupture [6]. Indications for endovascular or open elective repair are based upon a maximum AAA diameter and an ultrasound based AAA screening program for 65-year-old men is a highly cost-effective preventive health measure [7]. These programs, together with the growing use of abdominal imaging for other indications, have led to better identification of patients with small AAA. Still, the benefit of early diagnosis is limited because most detected AAA are below the currently accepted threshold for elective endovascular or surgical repair. Although the management of patients with small AAAs has recently been defined in the clinical practice guidelines [1], predicting progression of small AAAs in an individual patient is still difficult. The diameter of aneurysm may not be the only criterion for estimating growth, and a better understanding of biological mechanisms could help us to identify the mechanisms of drug therapy for slowing down growth [8].

Development and progression of AAA are connected with oxidative stress, aortic wall inflammation, elastin and collagen degradation, apoptosis and loss of extracellular matrix, neovascularization, and depletion of smooth muscle cells. Aortic dilation and rupture are probably caused by increased turnover and loss of fibrillary collagen and increased expression of collagenase, elastase, and matrix metalloproteinase [8]. Formation and progression of intraluminal thrombus may be involved in the evolution and possible rupture of AAA. Many AAAs show discontinuous growth patterns with periods of non-growth alternating with periods of acute expansion and rupture [9]. This observation may actually be a result of measurement error [10].

Surrogate biomarkers of development, growth and rupture could give precise information to guide proper management of AAA. New imaging modalities, such as volume measurement, biomechanical analysis, functional and molecular imaging, and circulating biomarkers, have shown a potential to be adopted in clinical practice in the future [11,12,13,14]. Development of novel high-throughput technologies (e.g., metabolomics) is a possible approach for a better understanding of the pathophysiology of AAA and for finding new possible biomarkers of AAA. Recently reported metabolomic changes in AAA patients were related to carbohydrate, lipid, and amino acid metabolism [15,16,17,18,19]. The main aim of the current study was to describe the profile of low molecular weight metabolites (<1 kDa) in AAA patients and to explore if low molecular weight metabolites are linked to AAA growth. 

## 2. Results

The clinical baseline characteristics of the infra-renal AAA patients and the controls are shown in Table 1. 

The patients of the AAA group had significantly older age and more comorbidities (hypertension, CAD, CVD), they were treated more frequently by acetylsalicylic acid (ASA) and statins, and there were more active smokers and less never-smokers among them compared with the control group. There were no significant differences in occurrence of diabetes or renal insufficiency between the study groups.

Out of 183 measured low-molecular weight metabolites, the levels of only four amino acids (histidine, asparagine, leucine, isoleucine) and four phosphatidylcholines (PC.ae.C34.3, PC.aa.C34.2, PC.ae.C38.0, lysoPC.a.C18.2) were found to be significantly reduced (*p* < 0.05) after adjustment for confounders (age, smoking status, hypertension, CAD, CVD, diabetes mellitus, renal insufficiency and medications (ASA, statins)) between the AAA patients (fast and slow growth rates combined) and the controls (Table 2, Figure 1). Appendix A contains the results of all 183 metabolites. There were no significant changes in the metabolites distinguishing the infra-renal AAA patients with slow or fast growth rate from the controls, or distinguishing the patients with low growth rate from those with high growth rate (data not shown). There were multiple significant linear correlations between eight metabolites (Table 3 and Table 4). Furthermore, after dividing AAA patients into terciles regarding aortic baseline diameter (36/41/47 mm), patients in third tercile (47 mm) had a negative correlation between PC.aa.C34.2 and aortic diameter (r = −0.50, *p* = 0.009); negative correlation also occurred between His and diabetes (r = −0.45, *p* = 0.02); His (r = −0.48, *p* = 0.01), Leu (r = −0.39, *p* = 0.04), PC.aa.C34.2 (r = −0.43, *p* = 0.03), PC.ae.C34.3 (r = −0.44, *p* = 0.02) and PC.ae.C38.0 (r = −0.39, *p* = 0.04) associated negatively with renal insufficiency; and PC.aa.C34.2 correlated negatively with usage of statins (r = −0.41, *p* = 0.04).

## 3. Discussion

The current study demonstrated several metabolomic shifts in male patients with infra-renal AAA compared to the aortic healthy controls. The main novel observation was that the patients with AAA had decreased levels of four amino acids and four phosphatidylcholines after adjustment for the potential confounders. Since no differences were identified between patients with slow and fast growing AAAs, the discussion will focus on the difference between patients with AAA and healthy controls. 

Relevant data about metabolomic alterations in AAA patients has been sufficiently investigated in a few articles [15,16,17,18,19]. There are some reports about the metabolomic profile in patients with thoracic aortic disease. Analyses made by Doppler et al. demonstrated a general increase in the aortic tissue’s total sphingomyelin levels in bicuspid aortic valve-associated thoracic aortic aneurysms and in tricuspid aortic valve-associated aortic dissections compared to controls [20]. It was demonstrated that sphingomyelins C16:0, C24:0, C16:1, and hydroxy-sphingomyelin C22:1 carried significant cardiovascular burden and their elevated levels were associated with the risk of myocardial infarction [21]. In the current study, one lysophosphatidylcoline (LysoPC) and three phosphatidylcholines (PC), but no sphingomyelin plasma levels, were significantly decreased in the AAA patients compared to the healthy controls. We used Biocrates AbsoluteIDQ^®^ p180 kit (Innsbruck, Austria), which does not enable separation of isomers of lipids; thus, for deeper understanding of the role of complex lipids in AAA pathogenesis we need further separate, well-focused, and standardized lipidomics studies [22]. Recently, it was also reported that sphingolipids and LysoPCs were significantly reduced in patients with acute aortic dissection [23]. An active sphingomyelinase-ceramide pathway, characterized by increased levels of ceramides and metabolisation of sphingomyelins (e.g., by oxidized LDL-induced sphingomyelinase activity), leads to reduced levels of sphingomyelins [24]. This pathway has pro-atherogenic, pro-oxidative, and pro-inflammatory activities, resulting in premature vascular ageing and cardiovascular events [25]. 

AAA and atherosclerosis have some similar pathophysiological processes, such as chronic inflammation, vascular smooth muscle cell apoptosis, extracellular matrix degradation, and thrombosis [26,27]. Although the two diseases share common risk factors, there is no proof that atherosclerosis and AAA have a causal relationship. Risk factors of atherosclerosis, i.e., increasing age, male gender, smoking, hypertension, and dyslipidaemia, are positively correlated with AAA [26,27,28]. High plasma lipoprotein a level is a risk factor for AAA [29], as lipoprotein a carries monocyte chemoattractant protein 1 and oxidized phospholipids, causing chronic inflammation, oxidative stress and injury of the arterial wall [27]. According to a clinical study, increased levels of arachidonic acid were related to AAA incidence and progression, and AAA patients with elevated arachidonic acid levels were more likely to require surgical repair [30]. Moreover, a previous clinical study reported that proprotein convertase subtilisin/Kexin type 9 (PCSK9) inhibitors, primary indicated for the treatment of hyperlipidemia, also reduce the risk of AAA [31].

In the current study, lysoPC.a.C18.2, PC.ae.C34.3, PC.aa.C34.2 and PC.ae.C38.0 were significantly decreased in the patients with AAA. This is consistent with a previous investigation of more than 3600 individuals from three population-based studies, which demonstrated that lysoPC.a.C18.2 were inversely associated with body mass index, markers of inflammation and subclinical cardiovascular disease, and moderately improved risk reclassification beyond traditional risk factors [32]. LysoPCs species, described further as being associated also with higher HDL-cholesterol and total cholesterol and lower BMI, mostly derive from phosphatidylcholines (PCs). Higher levels of pro-inflammatory and pro-atherogenic LysoPCs were noted during the oxidative modification of LDL-cholesterol that accompanies their conversion to atherogenic particles. However, as they are produced by the phospholipase A2 (PLA2)-like activity of Paraoxanase 1, LysoPCs contribute to inhibition of macrophage biosynthesis and consequently reduce cellular cholesterol accumulation and atherogenesis [33].

Recent reports have suggested a protective effect of LysoPCs on cardiovascular risk. In a study of type 2 diabetes, LysoPC 18:2 was found to be inversely associated with incident diabetes and impaired glucose tolerance [34]. Decreased levels of LysoPCs were also found in patients with aortic dissection [23]. Ciborowski et al. described lower LysoPCs concentrations in the plasma of patients with AAA compared to controls, with a clear trend for a decrease with increasing aneurysm size [17]. The authors suggested that the possible cause of the decrease in the amount of LysoPCs in the plasma of AAA patients depending on aneurysm size is their accumulation in intraluminal thrombus. A previous report showed an increased activity of PLA2 in the serum of AAA patients [35]. As the eicosanoids and lysophospholipids are highly important signalling molecules, the results of this study support Cibrowski’s hypothesis that increased activity of PLA2 could be a response of the AAA patients’ organism in order to augment the low level of lysophospholipids in their plasma. The finding of our study also indicates that AAA patients with lower PCs and LysoPCs may be exposed to vessel wall aneurysmal degeneration and increased risk of atherosclerotic cardiovascular disease.

In the current study, the patients with AAA were characterized by reduced levels of four amino acids. Histidine, an indispensable amino acid, is important for regulation of blood pH and for synthesis of haemoglobin, as well as responsible for general growth and natural repair processes. An important metabolic fate of histidine is to be a precursor for glutamic acid [35]. Therefore, reduced histidine level may occur in AAA patients due to reasons associated with the glutamic acid metabolic route. Glutamic acid is a biosynthetic donor of proline, which is a proteinogenic amino acid that is needed in large quantities for biosynthesis of collagen (e.g., in aortic wall) [36]. We hypothesize that lower histidine levels, and hence also decreased glutamic acid levels, express enhanced proline utilization to substitute for the amount of degraded collagen during aneurysm formation. 

The level of asparagine was lower than expected among AAA patients. An explanation of this reduction may be related to the decreased activity of plasma aspartic aminotransferase (AST) in patients with AAA [37]. Lower level of AST with elevated utilization of glutamic acid in the proline route is associated with the next metabolic shift: the level of aspartate may be reduced due to decreased transfer of the amino group from glutamic acid to oxaloacetate. Asparagine synthetase capability to use aspartate is therefore limited for asparagine production [38], and as a result, the level of asparagine declines. 

Recent research supports the notion that, in fact, elevations in essential branched-chain amino acids (BCAA) (e.g., leucine, isoleucine) contribute causally to insulin resistance, and these changes can predict development of diabetes [39]. Conversely, insulin increases whole-body BCAA oxidation, and inflammatory cytokines can double whole-body oxidation of BCAAs and their decrease [40]. In this study, decreased levels of leucine and isoleucine in the AAA patients supply further evidence of the inflammatory pathway in the pathogenesis of AAA. Lower BCAA levels may also lead to activation of beta-oxidation, which increases utilization of phosphatidylcolines, as supported by our finding about the reduced levels of lysoPC.a.C18.2, PC.ae.C34.3, PC.aa.C34.2, and PC.ae.C38.0. 

This pilot study requires validation, and the sample size was relatively small. We used plasma instead of aortic tissue for single time point sampling, of obvious reasons in the endovascular era, and also because we wanted to compare patients with slow and fast growth. Metabolomic shifts depend on the difference between the rates of low molecular metabolite biosynthesis and transition from tissue into the blood, as well as between the rates of their uptake and elimination from the blood. By focusing on tissue, one can exclude certain system-driven aspects in disease pathogenesis. Nevertheless, diseased aortic wall may produce systemic disease specific metabolic alterations that are reflected in plasma samples. Moreover, the approach of targeted metabolomics, which we opted for to find all analyzed metabolites, limited the results; therefore, associations between other metabolite classes might have been missed.

The current study identified multiple differences in the plasma metabolomics of AAA patients compared to aorta-healthy controls. Assessment of the levels of different low-molecular metabolites allows to improve the current understanding of the pathogenesis of AAA and to serve in the future metabolites as potential biomarkers.

## 4. Materials and Methods

### 4.1. Patient Cohort

Since 2008, all patients with AAA and age and gender matched healthy controls in Uppsala have been invited to donate blood to explore different biomarkers for AAA progression [41]. In the current study, blood was collected from two groups of patients with previously diagnosed AAA according to its progression: AAA with fast yearly growth rate (mean 3.3 mm; range 1.3–9.4 mm; *n* = 39) and with slow yearly growth rate (0.2 mm; range −2.6–1.1 mm; *n* = 40); as well as from healthy subjects (i.e., with non-aneurysmal aorta (*n* = 79). All aortic measurements in patients and controls were performed, using ultrasound, by registered nurses, specially trained in ultrasonography, or by ultrasound technicians. The maximum anteroposterior diameter of the infra-renal aorta was measured according to the leading-edge-to-leading-edge principle [10].

The median interval between the infra-renal aortic measurements was 3.8 years (range 1.3–14). The study was conducted according to the guidelines of the Declaration of Helsinki, and approved by the Research Ethics Review Board (EPN) of the Uppsala/Örebro region (Dnr 2007/052). Informed consent was given by all participants prior to the investigation.

The following inclusion criteria were used for AAAs; (1) aortic diameter ≥35 mm, (2) follow-up ≥6 months, and (3) ≤5 mm shrinkage during follow-up. In this pilot study, we included 40 patients with the slowest growth and 39 with the fastest growth. Additionally, 79 patients with a normal aorta at screening were selected as controls.

All participants were asked to complete a standardized health questionnaire on smoking habits and medical history. Coronary artery disease (CAD) was defined as a history of angina pectoris or myocardial infarction; cerebrovascular disease (CVD), as a history of stroke or transient ischemic attack (TIA); hypertension, as a history of hypertension or current antihypertensive medication; diabetes mellitus, as a history of dietary or medically treated diabetes; and renal insufficiency as a history of a clinically relevant renal impairment. Based on the smoking history, three groups were defined: never-smokers, former smokers, and active smokers. 

### 4.2. Sample Preparation

Acetonitrile, formic acid, and water (HPLC–grade) were purchased from Sigma-Aldrich (Germany). An Agilent Zorbax Eclipse (Agilent Technologies, Santa Clara, California, United States) XDB C18, 3.0 × 100 mm, 3.5 µm with Pre-Column SecurityGuard, Phenomenex (Phenomenex, Torrance, CA, USA), C18, 4 × 3 mm was used with the AbsoluteIDQ^®^ p180 kit (Biocrates Life Sciences AG, Innsbruck, Austria) for the targeted analysis of metabolites.

The serum samples were thawed on ice and processed following the steps in the AbsoluteIDQ^®^ p180 kit’s user manual. In summary, the serum (10 µL) was pipetted onto a 96-well plate with added internal standards and dried, using nitrogen, following a derivatization process using phenylisothiocyanate. All samples were measured on QTRAP 4500 (ABSciex, USA) coupled to Agilent 1260 series HPLC (USA), using the C18 column and flow injection analysis. The concentrations of the metabolites were calculated in the vendor’s software using internal standards’ intensities for reference.

### 4.3. Metabolomics Measurement

Biocrates’ commercially available AbsoluteIDQ^®^ p180 kit enables the quantification of up to 188 metabolites from different compound classes. The lipids and hexoses were measured by flow injection analysis-mass spectrometry (FIA-MS) and small molecules were measured by liquid chromatography-mass spectrometry (LC-MS). The experimental metabolomics measurement technique is described in detail by patents EP1897014B1 and EP1875401B1 (at https://patents.google.com/patent/EP1897014B1 (application granted 2014-01-15) and https://patents.google.com/patent/EP1875401B1 (application granted 2014-03-05)). Briefly, a 96-well based sample preparation device was used to quantitatively analyze the metabolite profile in the samples. This device consists of inserts that have been impregnated with internal standards, and a predefined sample amount was added to the inserts. Next, a phenyl isothiocyanate (PITC) solution was added to derivatize some of the analytes (e.g., amino acids). After the derivatization was completed, the target analytes were extracted with an organic solvent, followed by a dilution step. The obtained extracts were then analyzed with a LC-MS system. Concentrations were calculated using appropriate mass spectrometry software (LC part) and Biocrates MetIDQ™ software (MetIDQ Carbon version 6.0.0) (FIA part), and data were imported and merged in MetIDQ for further analysis.

The Biocrates AbsoluteIDQ^®^ p180 kit used in this study is standardized and quality controlled by the manufacturer. It has been proven to be a quantitative and reproducible solution for the measurement of various metabolites as demonstrated by an international ring trial [42], thus illustrating the precision and reproducibility of the analysis.

### 4.4. Statistical Analysis

Data were analyzed using the software R, version 3.3.1 for Windows. Continuous variables are expressed as mean or median and standard deviation (SD) or range and categorical data are expressed as the number (%) of patients or healthy subjects. The chi-square test was used to determine an association between categorical variables. To account for the differences in several clinical parameters between the groups, generalized linear models were built where clinical parameters were added as covariates. The Spearman rank correlation was performed to find potential bivariate associations. All *p*-values from the models were adjusted for multiple comparisons using a Benjamini–Hochberg false rate discovery of 5%. *p*-values < 0.05 were considered statistically significant.

## Figures and Tables

**Figure 1 metabolites-11-00555-f001:**
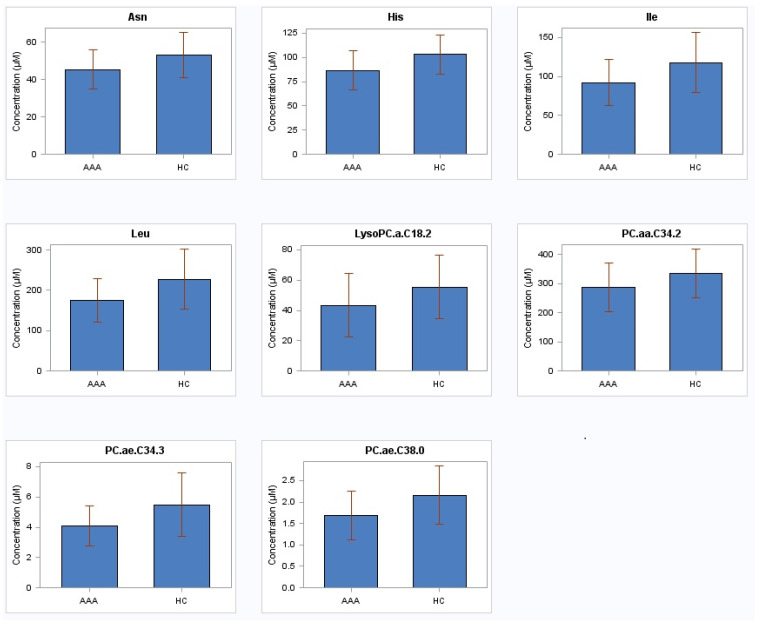
Amino acids and phosphatidylcholines in plasma from 79 patients with abdominal aortic aneurysm and 79 healthy subjects with non-aneurysmal aorta. AAA abdominal aortic aneurysm; HC healthy controls; His histidine; Asn asparagine; Leu leucine; Ile isoleucine; PC.ae.C34.3 phosphatidylcholine acyl-alkyl C34:3; PC.aa.C34.2 phosphatidylcholine diacyl C34:2; PC.ae.C38.0 phosphatidylcholine acyl-alkyl C38:0; LysoPC.a.C18.2 lysophosphatidylcholine acyl C18:2.

**Table 1 metabolites-11-00555-t001:** Baseline characteristics of the study population.

	Controls (*n* = 79)	AAA (*n* = 79)	*p*-Value
Baseline diameter, mm (95% CI)	19 (18–19)	42 (40–43)	<0.001
Age, years	All 65	68 (67–69)	0.01
Male gender	All men	All men	-
Comorbidities,			
Hypertension, % (95% CI)	32.9% (22.7–44.4)	67.5% (56.1–77.6)	<0.001
CAD, % (95% CI)	13.9% (7.2–23.6)	50.0% (38.6–61.4)	<0.001
CVD, % (95% CI)	3.8% (0.8–10.7)	18.8% (10.2–27.3)	0.04
Diabetes mellitus, % (95% CI)	6.3% (2.1–14.2)	15.0% (8.0–24.7)	0.07
Renal insufficiency, % (95% CI)	0%	10.0% (4.4–18.8)	0.06
Smoking,			
Never smoked, %	41.80%	16.50%	
Stopped smoking, %	55.70%	65.80%	<0.001
Active smoking, %	2.50%	17.70%	
Medication,			
ASA, % (95% CI)	24.1% (15.1–35.0)	58.2% (46.6–69.2)	<0.001
Statins, % (95% CI)	22.8% (14.1–33.6)	51.9% (40.3–63.3)	<0.001

Data is given as mean or percentage (95% confidence interval (CI)); AAA abdominal aortic aneurysm; CAD coronary artery disease; CVD cerebrovascular disease, ASA acetylsalicylic acid.

**Table 2 metabolites-11-00555-t002:** Low-molecular weight metabolites in plasma from infra-renal AAA patients and controls *.

Metabolites Amino Acids	Controls (*n* = 79)	All AAA (*n* = 79)	*p*-Value **	*p*-Value ***
His	103.0 ± 20.3	86.4 ± 20.1	0.018	<0.001
Asn	53.2 ± 12.2	45.5 ± 10.5	0.018	<0.001
Ile	118.0 ± 38.9	92.3 ± 29.5	0.043	0.001
Leu	227.7 ± 75.1	175.1 ± 55.0	0.043	0.001
**Phosphatidylcholines**				
PC.ae.C34.3	5.5 ± 2.1	4.1 ± 1.3	0.018	<0.001
PC.ae.C38.0	2.2 ± 0.7	1.7 ± 0.6	0.046	0.002
LysoPC.a.C18.2	55.5 ± 21.1	43.3 ± 20.8	0.046	0.002
PC.aa.C34.2	334.7 ± 84.0	286.8 ± 82.8	0.047	0.002

* Only metabolites that had significantly different levels after adjustment for clinical parameters (age, hypertension, coronary artery and cerebrovascular diseases, renal insufficiency, smoking, diabetes and medications (statins, acetylsalicylic acid)) comparing healthy subjects and AAA patients are given in this Table. Concentrations of all metabolites are presented as µM ± standard deviation. ** Benjamini–Hochberg adjusted false discovery rate (FDR) *p*-value, *** *p*-value from model. AAA abdominal aortic aneurysm; His histidine; Asn asparagine; Leu leucine; Ile isoleucine; PC.ae.C34.3 phosphatidylcholine acyl-alkyl C34:3; PC.aa.C34.2 phosphatidylcholine diacyl C34:2; PC.ae.C38.0 phosphatidylcholine acyl-alkyl C38:0; LysoPC.a.C18.2 lysophosphatidylcholine acyl C18:2.

**Table 3 metabolites-11-00555-t003:** Correlation matrix of metabolites of AAA patients (with r- and *p*-values).

	His	Asn	Ile	Leu	PC.ae. C34.3	PC.ae. C38.0	LysoPC.a. C18.2	PC.aa. C34.2
**His**	1.0	0.61 <0.001	0.51 <0.001	0.56 <0.001	0.52 <0.001	0.44 <0.001	0.58 <0.001	0.61 <0.001
**Asn**	0.61 <0.001	1.0	0.51 <0.001	0.54 <0.001	0.27 0.015	0.23 0.039	0.42 <0.001	0.23 0.044
**Ile**	0.51 <0.001	0.51 <0.001	1.0	0.94 <0.001	0.31 0.006	0.23 0.043	0.47 <0.001	0.42 <0.001
**Leu**	0.56 <0.001	0.54 <0.001	0.94 <0.001	1.0	0.28 0.011	0.26 0.021	0.39 <0.001	0.39 <0.001
**PC.ae.** **C34.3**	0.52 <0.001	0.27 0.015	0.31 0.006	0.28 0.011	1.0	0.53 <0.001	0.63 <0.001	0.77 <0.001
**PC.ae.** **C38.0**	0.44 <0.001	0.23 0.039	0.23 0.043	0.26 0.021	0.53 <0.001	1.0	0.44 <0.001	0.58 <0.001
**LysoPC.a.** **C18.2**	0.58 <0.001	0.42 <0.001	0.47 <0.001	0.39 <0.001	0.63 <0.001	0.44 <0.001	1.0	0.66 <0.001
**PC.aa.** **C34.2**	0.61 <0.001	0.23 0.044	0.42 <0.001	0.39 <0.001	0.77 <0.001	0.58 <0.001	0.66 <0.001	1.0

AAA abdominal aortic aneurysm; His histidine; Asn asparagine; Leu leucine; Ile isoleucine; PC.ae.C34.3 phosphatidylcholine acyl-alkyl C34:3; PC.aa.C34.2 phosphatidylcholine diacyl C34:2; PC.ae.C38.0 phosphatidylcholine acyl-alkyl C38:0; LysoPC.a.C18.2 lysophosphatidylcholine acyl C18:2.

**Table 4 metabolites-11-00555-t004:** Correlation matrix of metabolites of controls (with r- and *p*-values).

	His	Asn	Ile	Leu	PC.ae. C34.3	PC.ae. C38.0	LysoPC.a. C18.2	PC.aa. C34.2
**His**	1.0	0.71 <0.001	0.58 <0.001	0.59 <0.001	0.47 <0.001	0.38 <0.001	0.41 <0.001	0.50 <0.001
**Asn**	0.71 <0.001	1.0	0.41 <0.001	0.37 <0.001	0.45 <0.001	0.29 0.008	0.33 0.003	0.32 0.004
**Ile**	0.58 <0.001	0.41 <0.001	1.0	0.97 <0.001	0.25 0.03	0.30 0.008	0.19 0.09	0.46 <0.001
**Leu**	0.59 <0.001	0.37 <0.001	0.97 <0.001	1.0	0.22 0.05	0.28 0.01	0.14 0.23	0.44 <0.001
**PC.ae.** **C34.3**	0.47 <0.001	0.45 <0.001	0.25 0.03	0.21 0.05	1.0	0.51 <0.001	0.65 <0.001	0.67 <0.001
**PC.ae.** **C38.0**	0.38 <0.001	0.29 0.008	0.30 0.008	0.28 0.014	0.51 <0.001	1.0	0.40 <0.001	0.57 <0.001
**LysoPC.a.** **C18.2**	0.41 <0.001	0.33 0.003	0.19 0.09	0.14 0.23	0.65 <0.001	0.40 <0.001	1.0	0.57 <0.001
**PC.aa.** **C34.2**	0.50 <0.001	0.32 0.004	0.46 <0.001	0.44 <0.001	0.67 <0.001	0.57 <0.001	0.57 <0.001	1.0

His histidine; Asn asparagine; Leu leucine; Ile isoleucine; PC.ae.C34.3 phosphatidylcholine acyl-alkyl C34:3; PC.aa.C34.2 phosphatidylcholine diacyl C34:2; PC.ae.C38.0 phosphatidylcholine acyl-alkyl C38:0; LysoPC.a.C18.2 lysophosphatidylcholine acyl C18:2.

## Data Availability

The data presented in this study are available in article and Appendix A.

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
