# Peer review of "Metabolomic Profile of Abdominal Aortic Aneurysm"

_metabolites, 2021, doi:10.3390/metabo11080555_

Round 1
Reviewer 1 Report
Congratulations.
It is nice presentattion of metabolomic profile of abdominal aortic aneurysm.
The graph presentation and tables are sufficient and valuable.,
I have two minor concerns:
- What kind of examination of aneurysm growth? CT or USG
- It is accepted to present material and methods before results.
- It is kinfd of pilot study - I would include it in limitations section.
Author Response
Review Response
Manuscript (metabolites-1307989) entitled "Metabolomic profile of abdominal aortic aneurysm“
Comments of the Reviewer #1:
Congratulations. It is nice presentation of metabolomic profile of abdominal aortic aneurysm. The graph presentation and tables are sufficient and valuable. I have two minor concerns:
What kind of examination of aneurysm growth? CT or USG
Author’s Reply:
The aortic measurements were performed with ultrasound. Accordingly, the clarified and more detailed passage text now reads as follows (Page 10, section MATERIALS AND METHODS, lines 261-264):
All aortic measurements in patients and controls were performed, using ultrasound, by registered nurses, specially trained in ultrasonography, or by ultrasound technicians. The maximum anteroposterior diameter of the infra-renal aorta was measured according to the leading-edge-to-leading-edge principle [41].
Added reference:
- Gurtelschmid, M.; Björck, M.; Wanhainen, A. Comparison of three ultrasound methods of measuring the diameter of the abdominal aorta. Br. J. Surg. 2014,101,633-636.
It is accepted to present material and methods before results.
Author’s Reply:
We followed the example of the journal`s style, where the Material and Methods section are after the Discussion paragraph.
It is kind of pilot study - I would include it in limitations section.
Author’s Reply:
This is true and we changed the manuscript according to your suggestion. The modified text now reads as follows (Page 9, section DISCUSSION, line 238):
This pilot study requires validation, and the sample size was relatively small.

Reviewer 2 Report
This study was mainly based on the targeted analysis of 180 metabolites in plasma isolated from AAA patients and aorta-healthy controls to explore if low molecular weight metabolites are linked to AAA growth.. The authors identified levels of only four amino acids (histidine, asparagine, leucine, isoleucine) and four phosphatidylcholines (PC.ae.C34.3, PC.aa.C34.2, PC.ae.C38.0, lysoPC.a.C18.2) to be downregulated in the AAA patients compared with the controls after adjusting for confounding risk factors. Ideally such an assessment is highly relevant when performed on AAA tissues, however as open repairs are increasingly rate these days, the authors justifies their findings based on available samples to test.
The finding of the study is significant that low molecular weight metabolites identified could be crucial in AAA. However as there was no significance between slow and fast growing AAAs, the relevance of the finding with the pathology and mechanism underlying the progression of the disease is unclear.
The authors suggested that Sperman's correlation test was performed. However, no data assessing the correlation between the significant metabolites and AA diameter is provided.
Author Response
Review Response
Manuscript (metabolites-1307989) entitled "Metabolomic profile of abdominal aortic aneurysm“
Comments of the Reviewer #2:
This study was mainly based on the targeted analysis of 180 metabolites in plasma isolated from AAA patients and aorta-healthy controls to explore if low molecular weight metabolites are linked to AAA growth. The authors identified levels of only four amino acids (histidine, asparagine, leucine, isoleucine) and four phosphatidylcholines (PC.ae.C34.3, PC.aa.C34.2, PC.ae.C38.0, lysoPC.a.C18.2) to be downregulated in the AAA patients compared with the controls after adjusting for confounding risk factors.
Ideally such an assessment is highly relevant when performed on AAA tissues, however as open repairs are decreasingly rate these days, the authors justifies their findings based on available samples to test.
The finding of the study is significant that low molecular weight metabolites identified could be crucial in AAA. However as there was no significance between slow and fast growing AAAs, the relevance of the finding with the pathology and mechanism underlying the progression of the disease is unclear.
Author’s Reply:
We agree that information about aortic tissue metabolomics is very interesting and, in comparison with serum metabolomics, it is highly relevant. However, the study patients` aneurysms were below the treatment criteria, and therefore their aortic tissue was unavailable.
Recently were reported several metabolomics shifts in AAA patients; so the main aim of the current study was to describe the profile of low molecular weight metabolites in AAA patients and to explore if low molecular weight metabolites are linked to AAA growth. Since no differences were found between patients with slow and fast growing AAAs, the discussion about findings in the manuscript only focuses on the difference between the patients with AAA and the healthy controls. We agree that information about the mechanisms (and related biomarkers) underlying the progression of the disease is clinically relevant and needs further exploration in future studies.
The authors suggested that Sperman's correlation test was performed. However, no data assessing the correlation between the significant metabolites and AAA diameter is provided.
Author’s Reply:
In addition to evaluation of potential associations between metabolite levels and AAA growth rate, we assessed also associations between other clinically relevant characteristics (e.g. size of AAA) and different metabolites (e.g. amino acids, phosphatidylcholine etc levels), but there were no significant associations/correlations when comparing patients and controls.
We changed the manuscript according to your suggestion and added data about the the investigated associations to the text.
The modified text now reads as follows (Page 5, section RESULTS, lines 122-124): No significant associations were detected in metabolites and clinical parameters (e.g. aortic diameter, age, comorbidities etc) when comparing patients and controls (data not shown).

Reviewer 3 Report
This manuscript is well written with clear hypothesis, appropriate experimental design and logical data presentation. I have two comments to the manuscript. The authors showed there is no association with the rate of AAA growth, but do the changes in amino acid and phosphatidylcholine levels correlated to the size of the AAA? Also, many markers/metabolites for AAA is overlapped with atherosclerosis, this should be discussed in the discussion.
Author Response
Review Response
Manuscript (metabolites-1307989) entitled "Metabolomic profile of abdominal aortic aneurysm“
Comments of the Reviewer #3:
This manuscript is well written with clear hypothesis, appropriate experimental design and logical data presentation. I have two comments to the manuscript.
The authors showed there is no association with the rate of AAA growth, but do the changes in amino acid and phosphatidylcholine levels correlated to the size of the AAA?
Author’s Reply:
In addition to evaluation of potential associations between metabolite levels and AAA growth rate, we assessed also associations between other clinically relevant characteristics (e.g. size of AAA) and different metabolites (e.g. amino acids, phosphatidylcholine etc levels), but there were no significant associations/correlations when comparing patients and controls.
We changed the manuscript according to your suggestion and added data about the the investigated associations to the text.
The modified text now reads as follows (Page 5, section RESULTS, lines 122-124): No significant associations were detected in metabolites and clinical parameters (e.g. aortic diameter, age, comorbidities etc) when comparing patients and controls (data not shown).
Also, many markers/metabolites for AAA is overlapped with atherosclerosis, this should be discussed in the discussion.
Author’s Reply:
We absolutely agree that AAA and atherosclerosis are overlapping diseases regarding their risk factors, pathophysiology and biomarkers. Accordingly, we changed the manuscript and added the most recent general data about this issue together with relevant references. We believe that this amendment puts our findings into the general context of the research field, thank-you!
The modified text now reads as follows (Page 7-8, section DISCUSSION, lines 171-183):
AAA and atherosclerosis have some similar pathophysiological processes, such as chronic inflammation, vascular smooth muscle cell apoptosis, extracellular matrix degradation, and thrombosis [25,26]. Although the two diseases share common risk factors, there is no proof that atherosclerosis and AAA have a causal relationship. Risk factors of atherosclerosis, i.e. increasing age, male gender, smoking, hypertension and dyslipidaemia, are positively correlated with AAA [25-27]. High plasma lipoprotein a level is a risk factor for AAA [28], as lipoprotein a carries monocyte chemoattractant protein 1 and oxidized phospholipids, causing therefore chronic inflammation, oxidative stress and injury of the arterial wall [26]. According to a clinical study, increased levels of arachidonic acid were related to AAA incidence and progression, and AAA patients with elevated arachidonic acid levels were more likely to require surgical repair [29]. Moreover, a previous clinical study reported that proprotein convertase subtilisin/Kexin type 9 (PCSK9) inhibitors, primary indicated for the treatment of hyperlipidemia, also reduce the risk of AAA [30].
(Page 8, section DISCUSSION, lines 189-196):
LysoPCs species, described further as being associated also with higher HDL-cholesterol and total cholesterol and lower BMI, mostly derive from phosphatidylcholines (PCs). Higher levels of pro-inflammatory and pro-atherogenic LysoPCs were noted during the oxidative modification of LDL-cholesterol that accompanies their conversion to atherogenic particles. However, as they are produced by the phospholipase A2 (PLA2)-like activity of Paraoxanase 1, LysoPCs contribute to inhibition of macrophage biosynthesis and consequently reduce cellular cholesterol accumulation and atherogenesis [32].
Added references:
- Toghill, B.J.; Saratzis, A.; Bown, M.J. Abdominal aortic aneurysm-an independent disease to atherosclerosis? Pathol. 2017,27,71-75.
- Hou, Y.; Guo, W.; Fan, T.; Li, B.; Ge, W.; Gao, R.; Wang, J. Advanced researchof abdominal aortic aneurysms on metabolism. Cardiovasc. Med. 2021,8,630269.
- Wanhainen, A.; Bergqvist, D.; Boman, K.; Nilsson, T.K.; Rutegård, J.; Björck, M. Risk factors associated with abdominal aortic aneurysm: a population-based study with historical and current data. Vasc. Surg. 2005,41,390–396.
- Kotani, K.; Sahebkar, A.; Serban, M.C.; Ursoniu, S.; Mikhailidis, D.P.; Mariscalco, G.; et al. Lipoprotein(a) levels in patients with abdominal aortic aneurysm. Angiology 2017, 68,99–108.
- Lindholt, J.S.; Kristensen, K.L.; Burillo, E.; Martinez-Lopez, D.; Calvo, C.; Ros, E.; et al. Arachidonic Acid, but not omega-3 index, relates to the prevalence and progression of abdominal aortic aneurysm in a population-based study of Danish men. Am. Heart Assoc. 2018,7,e007790.
- Klarin, D.; Damrauer, S.M.; Cho, K.; Sun, Y.V.; Teslovich, T.M.; Honerlaw, J.; et al. Genetics of blood lipids among ~300,000 multi-ethnic participants of the Million Veteran Program. Genet. 2018,50,:1514–1523.
- Rozenberg, O.; Shih, D.M.; Aviram, M. Human serum paraoxonase 1 decreases macrophage cholesterol biosynthesis: possible role for its phospholipase-A2-like activity and lysophosphatidylcholine formation. Thromb. Vasc. Biol. 2003,23,461-467.
